# Oncolytic Virus Encoding a Master Pro-Inflammatory Cytokine Interleukin 12 in Cancer Immunotherapy

**DOI:** 10.3390/cells9020400

**Published:** 2020-02-10

**Authors:** Hong-My Nguyen, Kirsten Guz-Montgomery, Dipongkor Saha

**Affiliations:** Department of Immunotherapeutics and Biotechnology, Texas Tech University Health Sciences Center School of Pharmacy, Abilene, TX 79601, USA; My.Nguyen@ttuhsc.edu (H.-M.N.); Kirsten.Montgomery@ttuhsc.edu (K.G.-M.)

**Keywords:** cancer immunotherapy, oncolytic virus, herpes simplex virus, immune checkpoint inhibitor, angiogenesis inhibitor

## Abstract

Oncolytic viruses (OVs) are genetically modified or naturally occurring viruses, which preferentially replicate in and kill cancer cells while sparing healthy cells, and induce anti-tumor immunity. OV-induced tumor immunity can be enhanced through viral expression of anti-tumor cytokines such as interleukin 12 (IL-12). IL-12 is a potent anti-cancer agent that promotes T-helper 1 (Th1) differentiation, facilitates T-cell-mediated killing of cancer cells, and inhibits tumor angiogenesis. Despite success in preclinical models, systemic IL-12 therapy is associated with significant toxicity in humans. Therefore, to utilize the therapeutic potential of IL-12 in OV-based cancer therapy, 25 different IL-12 expressing OVs (OV-IL12s) have been genetically engineered for local IL-12 production and tested preclinically in various cancer models. Among OV-IL12s, oncolytic herpes simplex virus encoding IL-12 (OHSV-IL12) is the furthest along in the clinic. IL-12 expression locally in the tumors avoids systemic toxicity while inducing an efficient anti-tumor immunity and synergizes with anti-angiogenic drugs or immunomodulators without compromising safety. Despite the rapidly rising interest, there are no current reviews on OV-IL12s that exploit their potential efficacy and safety to translate into human subjects. In this article, we will discuss safety, tumor-specificity, and anti-tumor immune/anti-angiogenic effects of OHSV-IL12 as mono- and combination-therapies. In addition to OHSV-IL12 viruses, we will also review other IL-12-expressing OVs and their application in cancer therapy.

## 1. Introduction

Interleukin 12 (IL-12) is a powerful master regulator of both innate and adaptive anti-tumor immune responses. As a heterodimeric cytokine, it produces multifaceted anti-tumor effects [1,2], including stimulation of growth and cytotoxic activity of natural killer (NK) cells and T cells (both CD4^+^ and CD8^+^) [1,3,4,5], induction of differentiation of CD4^+^ T cells towards Th1 phenotype [6,7], increased production of IFN-γ from NK and T cells [1,8,9], and inhibition of tumor angiogenesis [1,10]. Despite encouraging success in preclinical studies [4], the early stages of IL-12 clinical trials did not meet expectations. Severe adverse events were first reported on 15 out of 17 patients in a phase II clinical trial following intravenous IL-12 administration, and the trial was immediately terminated by the FDA following two cases of death [11,12]. Although success was observed in cutaneous T cell lymphoma variants [13,14], AIDS-related Kaposi sarcoma [15] and non–Hodgkin’s lymphoma [16], severity of side effects outweighed effectiveness of IL-12 based therapies in the vast majority of oncology clinical trials [17]. In an effort to optimize efficacy and enhance the safety profile, alternative approaches are being studied to localize IL-12 expression at the tumor microenvironment.

Recent studies show that systemic toxicity of IL-12 is limited when expressed by oncolytic viruses (OVs) locally in the tumors [18,19,20] and in the brains of non-human primates [21]. OVs are a distinct class of anti-cancer agents with unique mechanisms of action: (i) selectively replicating in and killing cancer cells (i.e., oncolysis) without harming healthy cells or tissues [22,23,24], and (ii) exposing viral/tumor antigens, which promote a cascade of anti-tumor innate and adaptive immune responses (i.e., in situ vaccine effects) [25,26]. The OV-induced vaccine effects can be further enhanced through viral expression of anti-tumor cytokines such as IL-12 [18,19,20], as illustrated in Figure 1. Cancer immunotherapy involving OVs is an emerging and increasingly examined therapeutic approach for the treatment of cancer [27,28]. Among OVs, oncolytic herpes simplex virus (OHSV) is the furthest along in the clinic and approved by the FDA for the treatment of advanced melanoma [29]. To utilize the therapeutic potential of IL-12, there are several OHSVs encoding IL-12 which have been genetically engineered and tested in various cancers (Table 1). In addition to OHSVs, several different OVs such as adenoviruses, measles virus, maraba virus, Newcastle disease virus, Semliki forest virus, vesicular stomatitis virus and Sindbis virus are also being engineered to express IL-12 (Table 2). Our literature research found that 25 different types of OV-IL12s are either being or have recently been explored (see Table 1 and Table 2). Despite this rapidly rising interest, there are no reviews on IL-12 expressing viruses that exploit their potential efficacy and safety to translate into human subjects. This review presents the most current data on this topic and provides a basic understanding of OV-IL12 as a promising treatment approach in cancer immunotherapy, which ultimately could support continued research in the future. More specifically, in this review, we will discuss safety, tumor-specificity, and anti-tumor/anti-vascular effects of OHSV-IL12 as monotherapy or combination therapy. In addition to OHSV-IL12 viruses, we will also review other IL-12 expressing OVs and their application in cancer therapy.

## 2. Genetically Engineered IL-12 Expressing OHSVs and Their Therapeutic Efficacy

### 2.1. IL-12 Insertion Does Not Compromise Safety and Tumor Specificity of Genetically Modified OHSV-IL12 Viruses

Herpes simplex virus type 1 (HSV-1) has a large 152 kb genome, and therefore, deletion or mutation of various genes (Figure 2) does not fundamentally alter its functional properties (such as infectivity, viral replication, etc.), but rather confers tumor cell-specific replication, with no or reduced toxicity (e.g., neuropathogenicity) [59]. For example, G47Δ-IL12 (an IL-12 expressing OHSV) has three genomic modifications that endow its tumor-specificity and safety: ICP6 inactivation, γ-(ICP)34.5 and ICP47 deletion, and IL-12 insertion [30,60]. ICP6, encodes for viral ribonucleotide reductase, controls nucleotide metabolism and helps HSV to replicate in healthy or non-dividing cells that are inherently lacking sufficient nucleotide pools [59,61] (Figure 3A). ICP6 inactivation by fusion of LacZ does not hamper DNA synthesis in cancer cells (Figure 3B) [59,62] but the lack of ribonucleotide reductase results in no nucleotide metabolism and no viral DNA replication in healthy cells (Figure 3C) [59,61,62,63]. Therefore, mutation in the ICP6 gene makes viral infection and replication tumor-specific and thereby increases safety. Similar to ICP6 inactivation, deletion of γ-34.5 also increases safety and cancer selectivity [22,59,64]. Healthy cells have various anti-viral defense mechanisms. For example, protein kinase R (PKR) phosphorylates eukaryotic translation initiation factor eIF2α, which shuts down synthesis of foreign proteins or viral antigens (Figure 4A) [65,66]. HSV with intact γ-34.5 overturns anti-viral defense in healthy cells through γ-34.5-mediated dephosphorylation of eIF2α and helps in viral protein synthesis and viral replication in healthy cells (Figure 4B) [67,68]. Therefore, γ-34.5 deletion results in no eIF2α dephosphorylation, and thereby, no protein synthesis and viral replication (Figure 4C) [67,68]. However, in cancer cells, γ-34.5-deleted HSV can freely replicate, since cancer cells usually have defects in anti-viral pathways such as PKR-eIF2α pathway (Figure 4D). ICP47 downregulates MHC class I presentation through inhibition of transporter-associated protein (TAP) channel [69,70] and prevents detection of OHSV-infected cancer cells by virus-specific CD8^+^ cells [71]. Thus, deletion of ICP47 enhances MHC class I expression and immune response against virus-infected tumor cells and the host’s anti-tumor immunity (innate and adaptive) [59,60,72]. ICP47 deletion also complements the loss of γ-34.5 through immediate early (IE) expression of unique short sequence US11 under the control of ICP47-IE promoter [59,73]. US11 is a true late gene that binds with PKR, preventing it from phosphorylating eIF2a [59,74,75]. IE expression of US11 keeps eIF2α dephosphorylated and helps viral protein translation and synthesis [59,74,75]. Finally, in order to improve efficacy, the master anti-tumor cytokine IL-12 has been inserted into the ICP6 region of OHSV G47Δ to create G47Δ-IL12 [30]. Viral expression of IL-12 does not compromise its tumor specificity and safety, but rather significantly improves its anti-tumor properties [19,26,30,76]. T-mfIL12 (another IL-12 expressing OHSV) has the same backbone as G47∆ but with mIL-12 inserted in the ICP6 deletion region [77]. In vivo safety data showed that intravenous application of T-mfIL12 (5 × 10^6^ pfu) in mice bearing subcutaneous, intracerebral or intravenously disseminated tumors is as safe as non-IL12 expressing OHSV [31].

Similar to OHSV G47Δ-IL12, a HSV-1/2 recombinant vaccine strain encoding IL-12 is constructed (designated NV1042), which has also multiple deletions/mutations: (i) deletion of one copy of ICP0, ICP4, ICP34.5, and one copy of UL*56* at the U_L/S_ junction, (ii) insertion of *Esherichia coli LacZ* gene under the control of the α47 promoter at the α47 locus, (iii) deletion of ICP47, and (iv) insertion of mIL-12 under the control of a hybrid a4-TK (thymidine kinase) promoter [32,59,78,79]. ICP0 is an important immediate early (IE) protein in switching viral lytic and latent phases that affects defense mechanisms of the host by blocking nuclear factor kappa B (NF-κB)-mediated transcription of immunomodulatory cytokines, inhibiting interferon regulatory factor 3 (IRF3) translocation to the nucleus, inhibiting gamma-interferon inducible protein 16 (IFI16), and degrading mature dendritic cell (DC) markers (CD83) [24,80]. After translocating to the host’s nucleus, ICP0 modulates different overlapping cellular pathways to regulate intrinsic and innate antiviral defense mechanism of host cells, allowing the virus to replicate and persist [80,81]. ICP4 blocks apoptosis and positively regulates many other genes in the HSV-1 genome necessary for viral growth [82]. Function of UL56 has not been fully studied but is thought to be involved in neuro-invasiveness of HSV-1 [78]. Therefore, removal of ICP0, ICP4, ICP34.5 and UL56 attenuates virulence and ensures selective viral replication in cancer. In vivo experiment shows no toxicity after intravenous administration of NV1042 (5 × 10^7^ pfu), as demonstrated by lack of cytopathic effects in vital organs (such as lung, brain, spleen, liver, and pancreas) during three months follow up [33]. However, its safety and tumor-selective replication is still a major concern especially for the treatment of tumors located in the central nervous system, since it has 1 intact copy of γ-34.5 (responsible for neuropathogenicity) and intact ribonucleotide reductase ICP6.

The OHSV M002 and M032 have deletion of both copies of γ-34.5, with murine and human IL-12 cDNA (p35 and p40 subunits, connected by an IRES), respectively, inserted into each of the γ-34.5 deleted regions [83,84,85,86]. M002 has been reported to be safe with no significant toxicity seen after intracerebral inoculation into mice or HSV-sensitive primate Aotus nancymae, despite long-term persistence of viral DNA [87]. M032, with demonstrated safety in non-human primates [21], is now in clinical trial in patients with recurrent glioblastoma (GBM) (see clinical section) [88].

Introducing multiple mutations or deletions in the OHSV genome to confer safety and cancer selectivity may lead to over-attenuation or undermine replication efficiency in cancer cells as opposed to its wild-type or lowly mutated/deleted HSV counterparts [38]. To address this issue, a recent next-generation retargeted IL-12-expressing OHSV known as R-115 has been developed. This OHSV contains no major mutation or deletion and expresses mouse IL-12 under a CMV promoter [38,89]. IL-12-armed R-115 is a derivative of R-LM113 [90]. R-LM113 is a recombinant human epidermal growth factor receptor 2 (HER2) retargeted OHSV with no IL-12 expression, and is successfully engineered by deleting amino acid residues 6 to 38 and by moving the site of single-chain antibody insertion in front of the nectin 1 interacting surface (i.e., at residue 39) [90]. Because of retargeting, it enters and spreads from cancer cell to cell solely via HER2 receptors, and has lost the ability to enter cells through natural glycoprotein D (gD) receptors, herpes virus entry mediator (HVEM) and nectin 1 [90]. Safety profile of R-115 is evaluated in immunocompetent (wt-C57BL/6) model and HER2-transgenic/tolerant counterparts. Mice receiving R-LM113 or R-115 resist very high intraperitoneal OHSV dose of 2x10^9^ PFU, which is a lethal dose for wild-type HSV that kills 83% animals [38]. In addition, 4 consecutive intratumoral injections of R-115 at 3–4 days interval shows no viral DNA in vital organs (blood, brain, heart, kidney, liver, brain and spleen) [38]. This indicates that IL12-armed R-115 is safe in mice. However, HER2 specificity makes R-115 applicable only in HER2-expressing tumors, such as mammary tumors, and merely suitable for the treatment of other lethal cancers, such as glioblastoma or other non-HER2-expressing tumors [90]. Recently, another IL-12-expressing OHSV MH1006 (ICP47 deletion and human IL-12 insertion) has been developed and found to be safe in an immunocompetent subcutaneous model of neuroblastoma [91]. However, MH1006 has an intact neurovirulence gene γ-34.5 and an intact ribonucleotide reductase ICP6, thus raising safety concerns in the brain [91]. In Table 1, we have listed all IL-12-expressing OHSVs with their genomic modifications and preclinical applications.

### 2.2. IL-12 Expressing OHSVs Produce Superior Anti-Tumor Immunity/Efficacy than Non-IL12 OHSVs

In an orthotopic, immunocompetent intracranial glioblastoma (GBM) stem-like cell model (005 GSC model), OHSV G47Δ-IL12 therapy provides superior results compared to OHSV G47Δ treatment alone. For example, G47Δ-IL12 causes significant extension of survival of mice compared with either G47Δ-empty (i.e., an OHSV with no IL-12 transgene expression; *p* < 0.005) or mock treatment (*p* < 0.001, >50% increase in median survival), with 10% of mice surviving long term [30]. Anti-tumor efficacy of G47Δ-IL12 is associated with a significant reduction of tumor cells (i.e., GFP^+^ 005 GBM stem-like cells or GSCs) and a robust immune alteration in the tumor, following single intra-tumoral virus injection [19]. The immune alteration mainly includes, but is not limited to, increased tumor infiltration of CD3^+^ and CD8^+^ T cells, reduction of immunosuppressive FoxP3^+^ regulatory T cells, and an increased ratio of CD8^+^ T cells/regulatory T cells (CD4^+^FoxP3^+^) (a hall mark of clinical efficacy) (Figure 5) [19,30]. The role of T cells in therapeutic efficacy is further investigated in athymic nude mice (i.e., T cell-deficient mice) bearing orthotopic brain tumors [30]. In the absence of T cells, G47Δ-IL12 treatment is unable to significantly enhance survival over G47Δ-empty treatment, indicating a critical role of T cells in the IL12-mediated anti-tumor activity [30]. While replicating in vivo in the tumor, G47Δ-IL12 treatment causes increased local production of IL-12 in the tumor, which is accompanied by a marked release of downstream Th1 mediator interferon gamma (IFN-γ) in the tumors and, to a lesser extent, in the blood [30]. IL-12/IFN-γ promotes differentiation of T cells towards Th1 phenotype [92], which further produces IFN-γ and anti-tumor immune effects, as opposed to Th2 type T cells [9,93]. Similarly, T-bet^+^ Th1 type cells are increased in the tumor following intra-tumoral G47Δ-IL12 treatment [19,26], though it has not been determined whether this increase is directly associated with OHSV-IL12-mediated IFN-γ production in the tumor. G47Δ-IL12 treatment promotes polarization of macrophages from pro-tumoral M2 towards anti-tumoral M1 (e.g., increased expression of iNOS^+^ and pSTAT1^+^ cells) without affecting total tumor-associated macrophage (TAM) population (Figure 5) [19], possibly because of IL-12 induced M1-polarizing IFN-γ expression [30]. G47Δ-IL12-mediated anti-cancer immune responses, i.e., in situ vaccine effect opens the door to combination treatment strategies involving other cancer immunotherapy drugs. In orthotopic malignant peripheral nerve sheath tumor (MPNST) models, a single intra-tumoral injection of G47Δ in sciatic nerve tumors, derived from human MPNST stem-like cells in athymic mice or mouse MPNST cells in immunocompetent mice, significantly inhibits tumor growth and prolongs survival, as compared to mock treatment [94]. Local IL-12 expression (i.e., G47Δ-IL12) further significantly improves the efficacy of G47Δ in an immunocompetent orthotopic MPNST model, indicating that IL-12 expression induces anti-MPNST immune responses and improves overall efficacy [94]. These studies support the application of G47Δ-IL12 in combination immunotherapies for MPNST tumors.

Similar to anti-tumor efficacy with G47Δ-IL12, NV1042 (i.e., another OHSV with IL-12 expression) treatment results in a striking reduction in squamous cell carcinoma (SCC) tumor volume compared with the tumors treated with NV1023 (i.e., OHSV lacking IL-12 expression) and NV1034 (i.e., OHSV lacking IL-12, but with GM-CSF expression) [32]. Fifty-seven percent of mice treated with NV1042 reject subsequent SCC re-challenge in the contralateral flank, indicating strong global anti-cancer immune response, as opposed to 14% mice treated with NV1023 or NV1034 [32]. Besides local application, NV1042 was intravenously administered for the treatment of spontaneous primary and metastatic prostate cancer in the transgenic TRAMP mice. Systemic IL12-expressing NV1042 was significantly more efficacious than non-IL12 expressing OHSV NV1023 in reducing the frequency of prostate cancer development and lung metastases [95]. NV1042 DNA was detected in primary and metastatic tumors at 2 weeks after the final systemic virus injection but not in liver or blood [95]. Similarly, anti-cancer efficacy of intravenously delivered NV1042 was also observed in disseminated pulmonary SCC. Compared to PBS and parental NV1023, the group treated with IL-12 expressing NV1042 completely showed no sign of pulmonary nodules at day 12. In a low tumor burden model, NV1042 treatment resulted in 100% survival, in contrast to 70% in NV1023-treated group and 0% in PBS-treated group [33]. Depletion of CD4^+^ and CD8^+^ T cells reduces anti-cancer efficacy of IL-12 expressing NV1042, which is similar to anti-cancer effects of non-IL12-expressing NV1023, indicating IL-12 expression plays an important role in enhancing oncolytic efficacy through immune modulation [33].

M002 treatment resulted in prolonged survival in both pediatric and adult intracranial patient-derived tumor xenograft models [34]. The better survival benefit is associated with OHSV receptor nectin-1 expression in tumor cells, which is usually higher in pediatric brain tumors than in adult GBMs [34]. In an immunocompetent breast cancer metastasis model, IL-12-armed M002 treatment significantly improved survival of mice over its parental unarmed OHSV R3659 (no IL-12 expression) [35], indicating IL-12 played a critical role for anti-tumor efficacy. In a syngeneic neuroblastoma model, single intracranial injection of M002 produced a minimal survival benefit over untreated mice [85], indicating the need for IL-12 in immunocompetent models. Similarly, mice bearing intracranial neuroblastoma treated intramuscularly (IM) with M002-infected irradiated neuroblastoma cells did not show any survival advantage over mice treated with non-infected irradiated tumor cells. However, a prime-boost vaccine strategy, such as IM injection of M002-infected irradiated tumor cells seven days prior to tumor implantation and seven days post-tumor implantation, produced sustained anti-tumor T-cell responses and significant survival advantage, as opposed to irradiated control tumor cells [85]. Because an important control group is missing in this experimental setup (i.e., unarmed OHSV-infected irradiated tumor cells), it is not clear whether this anti-cancer vaccine effect in neuroblastoma was due to OHSV, local IL-12 expression, or both. In syngeneic sarcoma models, M002 did not produce any survival benefit compared to its parental virus R3659 (no IL-12 expression) [96], despite M002 inducing a significant anti-tumor immune effect over R3659 treatment, such as an increased percentage of intra-tumoral CD8^+^ T cells and activated monocytes, a decreased percentage of myeloid-derived suppressor cells (MDSCs), and increased CD8:MDSC and CD8:T regulatory cell ratios [96]. In recently performed pilot experiments in an ovarian cancer metastatic model, systemic intraperitoneal application of M002 resulted in a robust tumor-antigen specific CD8^+^ T cell response in the peritoneal cavity and the omentum [97], which are the primary sites of ovarian cancer metastasis [98]. Because of the tumor-specific immunity, M002 treatment was more successful in controlling ovarian cancer metastasis and produced a significantly longer overall survival than mock treatment [97]. Whether the anti-tumor efficacy is minimal or better, local IL-12 expression (M002) creates a more favorable immune-active tumor microenvironment than unarmed OHSV, which makes tumors more responsive to other forms of immunotherapies, such as immune checkpoint blockades.

IL-12-armed R115 was superior in inducing local and systemic anti-tumor immunity and durable response over unarmed R-LM113 in both early and late schedules [38]. All mice that survived the primary tumor were protected from the distant tumor challenge and subsequent re-challenge. Treatment with R115 drove Th1 polarization, increased immunomodulatory cytokines such as IFN-γ, IL-2, Granzyme B, T-bet and TNF-α, and tumor infiltrating lymphocytes [38]. Tumor microenvironment of R115 group showed an increase in number of CD8^+^ and CD141^+^ cells, PD-L1^+^ tumor cells, and FoxP3^+^ T regulatory cells with a decrease in the number of CD11b^+^ cells [38]. In another study, a single R-115 injection in established tumors resulted in complete tumor eradication in about 30% of animals [39]. The treatment also induced a significant improvement in the overall median survival time of mice and a resistance to recurrence from the same neoplasia [39]. Interestingly, treatment with R-115 increases the number of CD4^+^ and CD8^+^ T cells infiltrating into the tumor microenvironment, while the vast majority of CD4^+^ and CD8^+^ T cells in the R-LM113 treatment group accumulated at the edge of the tumors, indicating the effects of IL-12 [39].

### 2.3. Anti-Tumor Anti-Vascular Effects of OHSV-IL12

IL-12 does not only enhances the anti-tumor immune effects of virotherapy, but it also suppresses the development of new blood vessels, a process termed angiogenesis [99], making it an anti-angiogenic cytokine. IL-12 elicits its anti-angiogenic effects through release of IFN-γ, which activates IFN-inducible protein 10 [IP-10 or CXC chemokine ligand (CXCL) 10], a chemokine that mediates chemotaxis of lymphocytes and angiostatic effects [10,17,100]. It has been demonstrated that IL-12-armed OHSV produces significantly higher level of anti-angiogenic effects (i.e., reduction of CD31^+^ blood vessels in the tumors) through IFN-γ/IP-10 pathway in an immunocompetent brain tumor model (Figure 5) [30], compared to non-IL12 OHSV. IL12-armed OHSV treatment also causes a reduction in vascular endothelial growth factor (VEGF) expression, another likely contributor in tumor angiogenesis (Figure 5) [30].

In a model of prostate cancer in transgenic TRAMP mice, treatment with IL-12 armed NV1042 significantly reduces expression of CD31^+^ vascularity compared to either NV1023 or mock treated tumors [95]. Anti-angiogenic property of NV1042 is confirmed by another study in a SCC model. Intratumoral delivery of NV1042 results in release of a high level of IL-12, as well as other secondary angiogenic mediators such as IFN-γ, monokine induced by gamma interferon (MIG), and IP-10. In contrast, IL-12 unarmed parental NV1023 treatment shows no increase in IL-12 expression and lower level of secondary angiogenic mediators [101]. These studies indicate that IL-12 gene transfer could significantly enhance unique anti-tumor and anti-angiogenic effects of virotherapy. These anti-angiogenic features allow OHSV-IL12 to be tested with other local or systemic angiogenesis inhibitors for an improved therapeutic outcome.

### 2.4. Inhibition of Tumor Angiogenesis Enhances Anti-Tumor Potential of OHSV-IL12 Treatment

Angiogenesis is one of the hallmarks of cancer. It plays a key role in cancer progression [102,103,104,105,106,107,108,109,110] and anti-angiogenic therapy has been an interesting target to control tumor growth [108,111,112]. Efforts to disrupt the vascular supply and starve the tumor from nutrients and oxygen have resulted in 11 anti-VEGF drugs approved for certain advanced cancers, either alone or in combination with chemotherapy or other targeted therapies. Unfortunately, this success has had only limited impact on overall survival of cancer patients, and rarely resulted in durable responses. Bevacizumab (Avastin), an FDA approved anti-angiogenic drug (anti-VEGF), did not show significant improvement in overall survival [112,113,114]. Therefore, other anti-angiogenic agents and combinatorial strategies are being tested to target complex tumor microenvironment.

Because OHSV G47Δ-IL12 does not only induce anti-tumor immunity but also produce anti-angiogenic activities [30], it is hypothesized that anti-tumor effects of G47Δ-IL12 treatment would synergize with anti-vascular drugs. Axitinib (AG-013736) is an FDA approved, orally administered potent small molecule tyrosine kinase inhibitor (TKI), which inhibits VEGF receptor (VEGFR) 1-3, platelet-derived growth factor receptor beta (PDGFR-β) and receptor tyrosine kinase c-KIT (CD117) [115], and shows promising anti-vascular and anti-tumor activity in a variety of advanced stage cancers, including GBM [116,117,118]. In addition to anti-vascular effects, it also induces anti-tumor immune effects [119,120]. Therefore, anti-vascular/immune axitinib was tested in combination with anti-angiogenic/immune stimulatory G47Δ-IL12 in highly angiogenic patient and mouse GSC-derived GBM models [76]. This combination significantly extends survival in both models and involves multifaceted anti-tumor activities including: direct oncolysis of tumor cells, extensive tumor apoptosis and necrosis, increased macrophage infiltration to the tumor, greatly reduced tumor vascularity (i.e., CD34^+^ blood vessels) and inhibition of angiogenic PDGFR/ERK pathway in patient GSC-derived GBM model, and T cell dependent activity in mouse GSC-derived GBM model (Figure 5) [76]. Since the anti-tumor efficacy of the dual combination therapy (G47Δ-IL12+axitinib) was T cell dependent, it is hypothesized that ICI (i.e., anti-PD-1 or anti-CTLA4) will improve the therapeutic outcome of G47Δ-IL12+axitinib dual combination. Interestingly, ICI did not improve anti-tumor effects of axitinib or G47Δ-IL12+axitinib combination therapy. This is in sharp contrast with the findings from other investigators, since they observed synergistic anti-tumor immune effects following axitinib + ICI combination therapy in preclinical syngeneic tumor models [121]. The underlying mechanism(s) behind the failure of ICI combination therapy in orthotopic brain tumor model is not clear and warrants further investigation. It is speculated that reduced vascular permeability after axitinib therapy may inhibit extravasation of T cells into the tumor [122]. Indeed, axitinib treatment significantly reduces T cell (CD3^+^ and CD8^+^) infiltration into brain tumors [76]. Because both axitinib and OHSV are already in clinical trials for brain tumors as monotherapy with limited efficacy, dual combination therapy (OHSV-IL12+axitinib) that shows anti-tumor efficacy in both immune deficient and immune competent orthotopic brain tumor models has translational relevance [76]. Because systemic anti-angiogenic therapy (i.e., axitinib) is often associated with renal toxicities [123,124], G47Δ-IL12 was tested in combination with a local OHSV expression of angiostatin (OHSV G47Δ-angio), an anti-angiogenic polypeptide, in hypervascular human GBM models [125]. The combination of two OHSVs (G47Δ-IL12+G47Δ-angio) significantly prolongs survival compare to each armed OHSV alone, which is associated with increased viral spread and reduced expression of VEGF and CD31^+^ blood vessels in the tumor (Figure 5) [125]. This study supports further engineering of OHSV to express both IL-12 and angiostatin locally in the tumor. Use of one virus rather than two is practical in the context of future FDA approval. Similar to G47Δ-IL12 and anti-angiogenesis studies, the combination of another IL-12 expressing OHSV NV1042 with the anti-cancer chemotherapy drug vinblastine results in significant reduction of tumor burden in athymic mice bearing subcutaneous CWR22 prostate tumors, which is most likely associated with diminishing the number of CD31^+^ endothelial cells [126].

### 2.5. Immune Checkpoint Inhibition Enhances OHSV-IL12 Treatment-Induced Anti-Tumor Immunity

Though anti-tumor effects of OHSV-IL12 therapy is multifaceted, virotherapy alone does not improve significant survival in cancer [19,30,76,127]. For example, G47Δ-IL12 monotherapy shows limited efficacy in preclinical immune competent models of prostate and malignant peripheral nerve sheath tumors [94]. Since OHSV-IL12 treatment changes immune phenotypes of the tumor microenvironment, it is tested in combination with other forms of immunotherapies (e.g., ICIs) to obtain a better therapeutic response [19,76]. ICIs, such as cytotoxic T lymphocyte antigen 4 (CTLA-4) and programmed death 1 (PD-1) suppress T cell-mediated anti-tumor immune responses (Figure 6), leading to tumor progression [128]. ICI antibodies (i.e., anti-CTLA-4 or anti-PD-1) are effective in unleashing tumor-induced immunosuppression and activating effector immune cells (Figure 6) [129]. Since OHSV-IL12 induces robust anti-tumor immunity [19], it is hypothesized that OHSV-IL12 will synergize with ICI antibodies and will improve the anti-cancer efficacy of G47Δ-IL12. Indeed, in 005 GSC-derived orthotopic brain tumor models, dual combination modestly extends survival compared to ICI antibody alone or G47Δ-IL12 therapy alone [19]. The modest anti-tumor efficacy of the dual combination is not due to the inability of ICI antibodies to cross the blood-brain barrier, since ICI antibodies were detected in the tumor [19]. Because CTLA-4 and PD-1 regulate anti-tumor immunity via distinct and non-redundant immune evasion mechanisms [130,131,132], it is hypothesized that combinatorial blockade of two immune inhibitory pathways will produce enhanced anti-tumor immune effects and will synergize with the anti-tumor efficacy of G47Δ-IL12 (i.e., triple combination therapy: G47Δ-IL12+anti-PD-1+anti-CTLA-4). Indeed, triple combination therapy leads to a significant percentage (89%) of long-term survivors (i.e., survived six months post-tumor implantation) [19]. These survivors remain protected following lethal tumor re-challenge in the contralateral hemisphere, surviving another three months until the experiment was terminated, which indicates development of long-term memory protection [19]. These unprecedented findings were reproduced in a second aggressive immune competent CT-2A GBM model [19]. The survival efficacy of the triple combination therapies is associated with a significant but complex immune alteration in the tumor microenvironment, as opposed to mock, single or dual combination therapies, which includes: i) increase tumor infiltration of T cells (CD3^+^ and CD8^+^); ii) increase number of proliferating T cells (CD3^+^Ki67^+^); iii) increase activated T cells (CD8^+^CD69^+^); iv) reduce regulatory T cells (FoxP3^+^); v) increase T effector (CD8^+^)/regulatory T cell (CD4^+^FoxP3^+^) ratio; vi) increase TAMs (CD68^+^, F4/80^+^); vii) skew TAMs towards M1-phenotypes (iNOS^+^, pSTAT1^+^, CD68^+^pSTAT1^+^); viii) increase Th1 differentiation (T-bet^+^); ix) reduce PD-L1^+^ cells; x) increase tumor-cell specific IFN-γ response; and (xi) reduce tumor cells (Figure 5) [19,26]. Depletion and inhibition of immune cell subtypes (i.e., CD4^+^ cell depletion by anti-CD4, CD8^+^ cell depletion by anti-CD8, peripheral macrophage depletion by liposomal clodronate, or CSF-1R inhibition by brain penetrant drug BLZ945 to target TAMs) confirms the necessity of CD4^+^ cells, CD8^+^ cells, and macrophages in the therapeutic efficacy, with CD4^+^ cells playing the critical role [19]. It remains to be determined how immune cells, especially CD4^+^ cells and M1-like macrophage polarization contribute to therapeutic efficacy. IL-12 appears to be critical for this exceptional anti-tumor efficacy, since another triple combination therapy involving the base G47Δ (without IL-12 expression) plus two systemic ICI antibodies results in only 13% long-term survivors [25], as opposed to 89% from G47Δ-IL12 + anti-PD-1 + anti-CTLA-4 combination [19].

## 3. Anti-Cancer Potential of other OVs Encoding IL-12

As IL-12 is a master anti-tumor cytokine and has distinct multifaceted anti-cancer properties [1,2,3,6,8,10], IL-12 expression by OVs is not limited to OHSVs. Several other OVs encoding IL-12 have been developed (Table 2), including adenoviruses [18,42,46,47,48,49,133,134,135,136,137,138], measles virus [20,50], maraba virus [51], Newcastle disease virus [52], Semliki forest virus [53,54,55,56,139,140,141], vesicular stomatitis virus [57,142], and Sindbis virus [58]. In these above-mentioned studies, IL-12 is expressed either alone [18,20,51,57,137,138,139] or co-expressed alongside with GM-CSF [134,135,136], pericellular matrix proteoglycan decorin [47], tumor necrosis factor-related apoptosis-inducing ligand (TRAIL) [133], IL-2 proinflammatory cytokine [52], IFN-γ inducing factor IL-18 [46], T cell co-stimulatory ligand 4-1BBL [49], CD8^+^ co-receptor for CD28 and CTLA-4 [48], or suicide genes [42]. The anti-tumor efficacy of these engineered OVs are tested in various mouse or hamster pre-clinical cancer models and produce superior anti-tumor immunity either alone [18,20,42,46,47,48,51,52,57,133,135,137] or in combination with other immunotherapeutic agents such as dendritic cell (DC) vaccine [49,134,136], ICI anti-PD-1 and anti-PD-L1 [139], or cytokine-induced killer cells [138].

The tumor-targeting oncolytic adenovirus (Ad-TD) delivers either wild-type IL-12 or non-secreting IL-12 (nsIL-12) directly to pancreatic tumor cells (Table 2) [44]. In a Syrian hamster tumor model, treatment with both Ad-TD-IL12 and Ad-TD-nsIL-12 results in 100% tumor eradication and animal survival, and treatment with Ad-TD-IL12 in particular produces a significant increase in populations of CD3^+^CD4^−^CD8^+^ in the spleen [44]. In addition, treatment with Ad-TD-nsIL-12 results in lower levels of lymph node IFN-γ, and splenic IFN-γ and IP-10 production [44]. Another oncolytic adenovirus expressing IL-12 (RdB/IL-12) inhibits tumor growth in murine melanoma lines by 95%, while adenovirus expressing both IL-12 and IL-18 (RdB/IL-12/IL-18) inhibits growth by 99% [46]. RdB/IL-12/IL-18 also increases the cytokine ratio of Th1/Th2, increases tumor infiltration of CD4^+^ T, CD8^+^ T and NK cells, and promotes differentiation of T cells expressing IL-12Rβ2 or IL-18Rα [46]. In another engineered oncolytic adenovirus (Ad-ΔB7/IL-12/4-1BBL), co-expression of both IL-12 and the cytokine 4-1BB ligand (4-1BBL) produces significantly higher survival in mice, with 100% of mice surviving more than 30 days after viral injection [49]. This is considerable when in comparison to the 20% survival rate of mice treated with virus expressing only IL-12 or 4-1BBL. In this study, mice treated with either Ad-ΔB7/IL-12 or Ad-ΔB7/IL-12/4-1BBL have a higher amount of tumor infiltrating CD4^+^ and CD8^+^ cells in comparison to treatment with Ad-ΔB7/4-1BBL or Ad-ΔB7 [49].

An oncolytic measles virus encoding an IL-12 fusion protein (MeVac FmIL-12) as a single agent produces potent anti-tumor immune effects in an immunocompetent colon cancer model with 90% complete remission [20]. This robust anti-cancer efficacy is dependent on T cells, particularly CD8^+^ cells, and is associated with activation of early NK cells and effector T cells, and upregulation of effector anti-tumor cytokines IFN-γ and TNF-α [20]. The findings are similar to what we observed in GBM with OHSV G47Δ-IL12 virus [19,30]. Although it was not examined whether the long-term survivors in MeVac FmIL-12 group developed any memory protection, the potent anti-tumor immune efficacy indicates that MeVac FmIL-12 is also an attractive candidate in cancer therapy [20]. Similar to the MeVac FmIL-12 virus, IL12-expressing oncolytic maraba virus (designated MG1-IL12-ICV) treatment also leads to complete eradication of peritoneal carcinomatosis, which is associated with significant recruitment of NK cells in the tumor microenvironment [51]. The lentogenic Newcastle disease virus Clone30 strain was generated to express IL-12 (designated rClone30-IL-12) that displays improved survival and reduced tumor volume [52]. In this case, co-expression of two cytokines (IL-12 and IL-2) produced synergistic effects on treatment, with the rClone30-IL-12-IL-2 virus inducing the greatest release of IFN-γ and IP-10 [52].

The Semliki Forest virus (SFV) has a broad host range and is suicidal, causing apoptosis in infected cells, which makes it a promising viral vector [53]. In a murine tumor model using B16 cells, a single intratumoral injection of SFV expressing IL-12 (SFV-IL12) results in significant tumor regression seven days after injection. This follows a distinct inhibition of tumor vascularization and an increase in IFN-γ production at two days post-injection [53]. Furthermore, treatment with SFV-IL12 results in an increase in expression of splenic IP-10 and monokine induced by interferon-γ (MIG) [53]. Similar results are obtained on the murine colon adenocarcinoma cell line, MC38, with SFV-IL12 treatment resulting in increased tumor-specific CD8^+^ T lymphocytes, reduced tumor volume, and improved survival [55]. Of particular notice, treatment of MC38 cells with SFV-IL12 results in tumor-infiltrating monocytic myeloid-derived suppressor cells (M-MDSCs) displaying increased expression of CD11c, CD8α, CD40, and CD86 in the presence of an intact, endogenous host type-I interferon (IFN-I) system [55].

Vesicular stomatitis virus (VSV) is another example of a viral vector [57]. Oncolytic VSV carrying IL-12 (rVSV-IL12) succeeds in effectively reducing tumor volume and prolonging survival in both human and murine SCC tumors, with 40% of mice treated with rVSV-IL12 surviving past 100 days post-injection, in comparison to mice treated with the fusogenic OV, rVSV-F, surviving only to 60 days post-injection (*p* < 0.0001) [57]. Similarly, treatment with the Sindbis viral vector carrying IL-12 (Sin/IL12) increases production of IFN-γ, and results in reduced tumor growth and improved survival in ovarian clear cell carcinoma [58]. Treatment with Sin/IL12 also modulates the regulatory functions of NK cells, increasing activation and recruitment of the cells [58]. These and afore-mentioned studies clearly suggest that IL-12 is a useful anti-cancer agent for oncolytic immunovirotherapy, and boosts anti-cancer immune properties of OVs.

## 4. Clinical Perspectives

Regardless of cancer types, IL-12 expressing OVs have been tested and found effective against various cancers (such as glioma, neuroblastoma, squamous cell carcinoma, metastatic breast cancer, hepatoblastoma, sarcoma, kidney cancer, lymphoma, prostate cancer, pancreatic cancer, colon cancer, ovarian cancer, melanoma, etc.) (Table 1 and Table 2). This clearly suggests that IL-12 expressing OVs are an attractive therapeutic candidate for clinical translation against any forms of cancer. In general, when translating results from bench to bed side, safety remains the most concerning aspect, along with dose, route of administration, viral pharmacokinetics and resistance mechanism of host cells [143]. Recent FDA approval of OHSV T-VEC in 2015 has heightened the field of oncolytic virus-based immunotherapy. The FDA approved OHSV expresses GM-CSF instead of IL-12. The safety and efficacy of T-VEC in immune-privileged organs, such as brain, has not been extensively elucidated. Moreover, T-VEC has not demonstrated durable responses in a majority of advanced melanoma patients [29], especially those with visceral metastases [144,145], raising questions about its possible long-term efficacy in patients with difficult-to-treat metastatic cancers.

Safety and anti-tumor efficacy of OHSV-IL12 as monotherapy or combination therapy has been demonstrated in preclinical tumor models [19,30,76]. Currently, G47Δ expressing human IL-12 is under development for clinical use. Safety of M002 has been established in the brain after intracerebral administration to non-human primates [21]. M032 is now in a phase I clinical trial (NCT02062827) in patients with recurrent/progressive GBM, anaplastic astrocytoma, or GBM [88]. Similar to OHSV-IL12, an adenovirus expressing human IL-12 is also under clinical trial investigation as monotherapy in prostate cancer (NCT02555397, NCT00406939), pancreatic cancer (NCT03281382), breast cancer (NCT00849459) and melanoma (NCT01397708). In addition, Ad-RTS-hIL-12 (another adenovirus encoding for IL-12) is under clinical trial evaluation in combination with veledimex (an oral activator ligand to promote release of IL-12 by an OV) in pediatric brain tumor (NCT03330197) and adult glioblastoma or malignant glioma (NCT02026271). Triple combination of Ad-RTS-hIL-12, veledimex and nivolumab (anti-PD-1) are also in active status in glioblastoma patients (NCT03636477). In our opinion, since OV-IL12 treatment induces prominent anti-tumor immunity including increased expression of PD-L1 in the tumor microenvironment, OV-IL12 therapy may improve the response rate to anti-PD-L1 treatment, especially in cancer patients who inherently lack PD-L1 expression and/or previously unresponsive to anti-PD-L1 treatment. Thus, combination studies involving OV-IL12 and ICI warrant clinical investigation and could be an attractive treatment strategy for cancer patients.

## 5. Conclusions and Future Directions

Genomic manipulation and understanding of pathogenicity have made OVs an attractive candidate for cancer therapy. Among OVs, OHSV is FDA approved for cancer treatment and is the furthest along in the clinic [29,144,145,146]. Moreover, the availability of antiviral drugs, such as acyclovir, makes OHSV a safer anti-cancer candidate over other OVs [59,147]. However, we all have recognized that OHSV expression of IL-12 and its application as a monotherapy does not provide a desired therapeutic outcome, as demonstrated in several preclinical cancer models [30,84,86,148,149]. It requires synergistic or additive combination approach with other anti-cancer therapies for an improved therapeutic outcome [19,25,76,148,150,151]. Similar to OHSV-IL12, a combination immunotherapeutic approach is also required to enhance anti-tumor efficacy of other OVs encoding IL-12 [49,134,136,138,139].

Although ICI-based cancer immunotherapy is rapidly evolving in the field of oncology, the combination therapy involving single or multiple ICIs is often associated with significant toxicity in humans [152,153,154,155,156]. Moreover, ICI immunotherapy has not been successful in devastating cancer types such as GBM (CheckMate-143) [157,158] or triple-negative breast cancer (KEYNOTE-119) [159]. In contrast, OV expressing single or multiple immune stimulator does not cause toxicity when expressed locally in the tumor by the virus [21,52,160,161,162,163], even in an immune-privileged brain [21], but rather induces robust anti-tumor immunity. Thus, developing appropriately designed and stronger version of viral vectors expressing multiple immune stimulators alongside the master anti-tumor cytokine IL-12 may induce superior local anti-tumor immune responses, while reducing the need for multiple systemic ICI or other systemic drugs, and eventually thwarting the current limitation of systemic cancer immunotherapy.

Besides considering construction of new viral vectors, another issue that needs to be addressed before utilizing the full potential of OVs is limited viral spread in tumors [19] due to presence of anti-viral genes or up-regulation of anti-viral factors following OV treatment [164]. Restricted viral replication and spread in the tumor may result in reduced tumor oncolysis with limited in situ vaccine effect. Identification, followed by inhibition of anti-viral factor(s) will provide tools to develop new OV-based immunotherapeutic strategies to enhance viral replication and spread in the tumor, and to induce potent anti-tumor immunity [164]. Designing a better rationale OV-based combination treatment strategy without compromising safety will be the key for clinical success.

## Figures and Tables

**Figure 1 cells-09-00400-f001:**
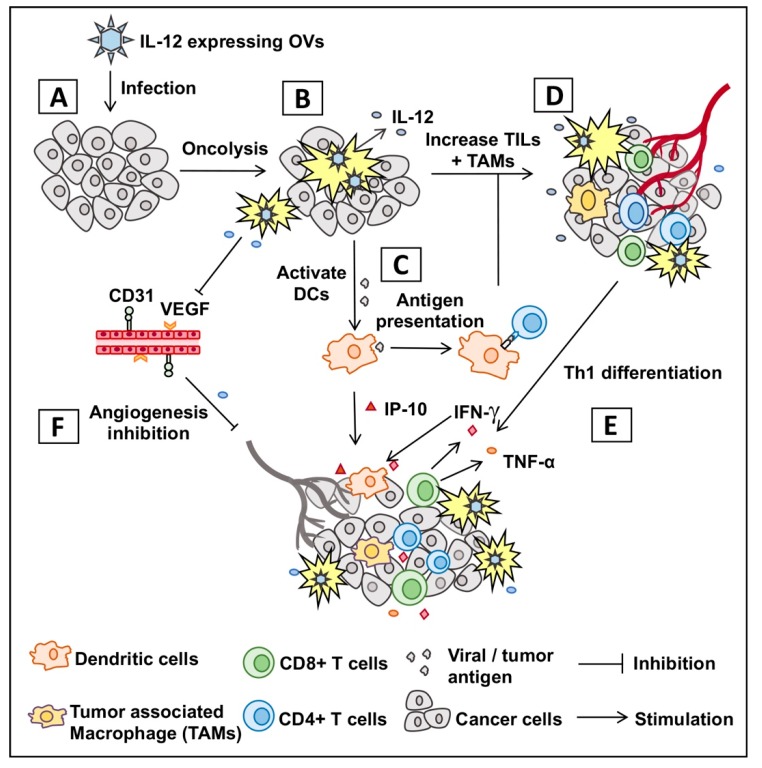
Graphic presentation of mechanism of action of oncolytic virus encoding IL-12. (**A**) Infection of tumor with oncolytic virus encoding IL-12 (OV-IL12). (**B**) OV-IL12 replicates in and kills cancer cells (i.e., oncolysis) and releases IL-12 in the tumor microenvironment. (**C**) Neoantigens from lysed cancer cells activate and recruit dendritic cells (DCs) into the tumor microenvironment. DCs process neoantigens, travel to nearest lymphoid organs, and present the antigen to T cells (CD4^+^ and CD8^+^ T cells). (**D**,**E**) T cells migrate to the site of infection (referred as tumor-infiltrating T cells or TILs), differentiate into Th1 cells, produce anti-tumor cytokines and kill cancer cells. (**F**) IL-12-induced production of IFN-γ and interferon inducible protein 10 (IP-10) produces anti-angiogenetic effect through reduction of tumoral vascular endothelial growth factor (VEGF) and CD31^+^ tumor endothelial cells.

**Figure 2 cells-09-00400-f002:**
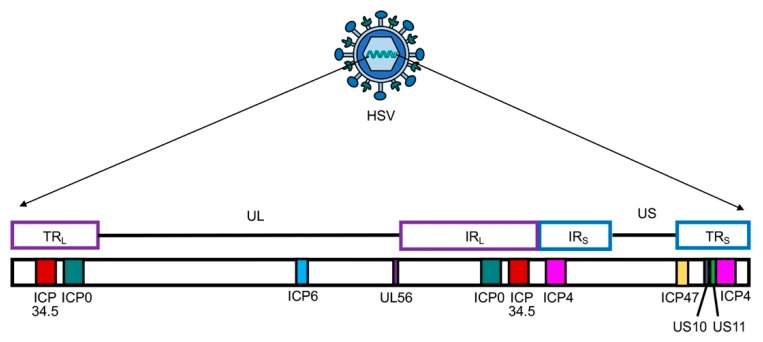
Schematic presentation of herpes simplex virus (HSV) genome with unique long (UL) and short (US) sequences. TR_L or S_—terminal repeat long or short; IR_L or S_—internal repeat long or short. Only genes that are modified and/or deleted during construction of OHSV-IL12 are presented. ICP, infected cell protein.

**Figure 3 cells-09-00400-f003:**
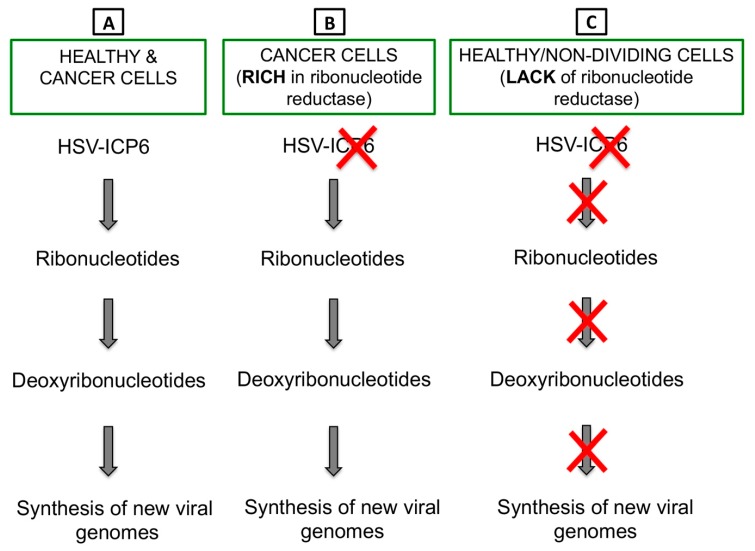
ICP6 inactivation drives tumor-specific replication of OHSV-IL12. (**A**) ICP6 encodes for large subunit of ribonucleotide reductase, which controls nucleotide metabolism and helps HSV to replicate in normal or healthy host cells that are inherently lacking or have insufficient nucleotide pools. (**B**) Cancer cells are rich in ribonucleotide reductase, thus HSV with an inactivated ICP6 does not hamper DNA synthesis in cancer cells. (**C**) Healthy or non-dividing cells lack ribonucleotide reductase. Thus, infection of healthy cells with an ICP6-inactivated HSV leads to no nucleotide metabolism and no viral DNA replication.

**Figure 4 cells-09-00400-f004:**
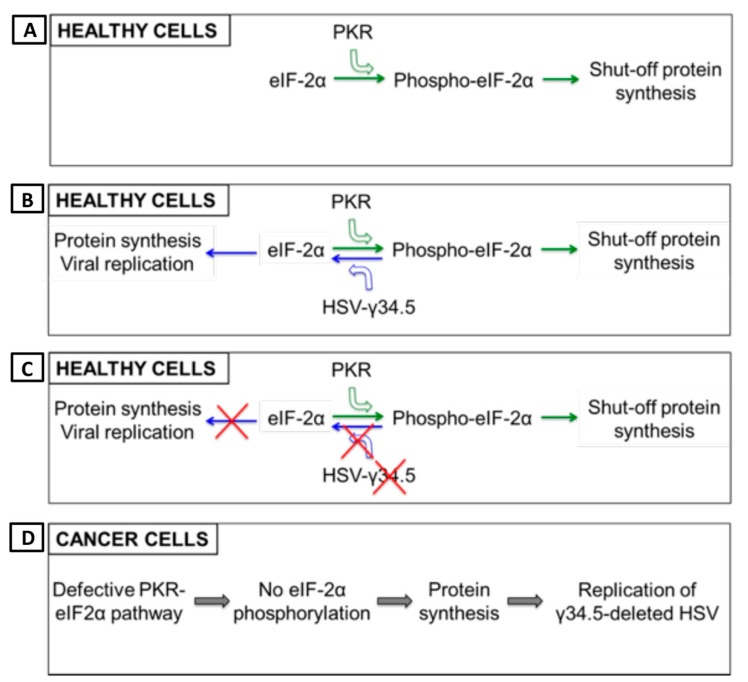
γ-34.5 deletion enhances safety and tumor-specificity of OHSV-IL12. (**A**) Healthy cells have inherent anti-viral defense mechanisms, such as protein kinase R (PKR). PKR phosphorylates translation initiation factor eIF2α, which shuts down synthesis of foreign proteins or viral antigens. (**B**) OHSV with an intact γ-34.5 overturns anti-viral defense in healthy cells through γ-34.5-mediated dephosphorylation of eIF2α and helps in viral protein synthesis/viral replication in healthy cells, leading to development of disease. (**C**) γ-34.5 deletion results in no eIF2α dephosphorylation in normal or healthy cells, and thereby, no protein synthesis and viral replication. (**D**) Cancer cells usually have defective PKR-eIF2α pathway, thus no inhibition of foreign protein synthesis. Therefore, γ-34.5-deleted OHSV can freely replicate in cancer cells.

**Figure 5 cells-09-00400-f005:**
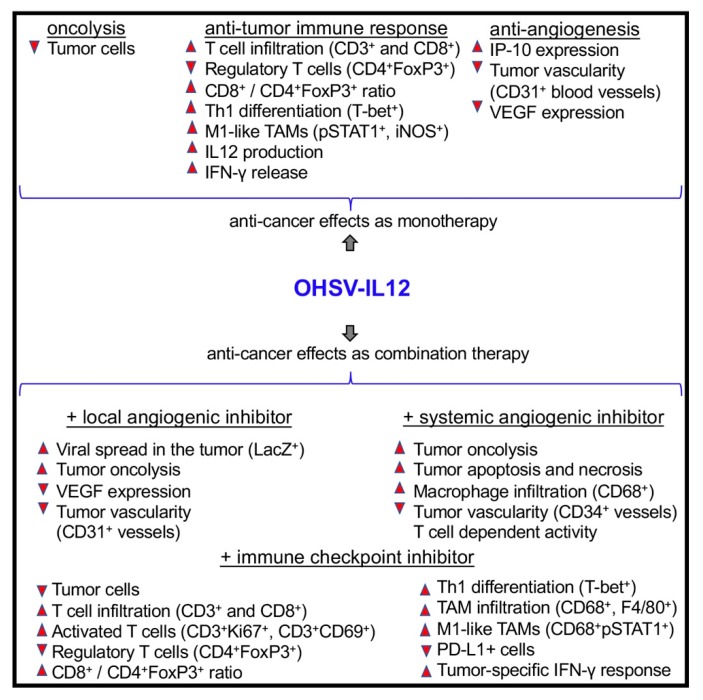
Anti-tumor effects of OHSV-IL12 treatment as mono- and combination-therapies. OHSV-IL12 treatment as monotherapy leads to three distinct anti-cancer effects: 1. Oncolysis, leading to reduction of cancer cells; 2. Induction of anti-tumor immunity, which is characterized by increased intratumoral infiltration of T cells, reduction of regulatory T cells, increased T effector/regulatory T cell ratio, enhanced Th1 differentiation, increased polarization of macrophages toward anti-tumoral M1-phenotype, and increased production of IL-12 and IFN-γ; and 3. Inhibition of tumor angiogenesis as demonstrated by reduced CD31^+^ blood vessels and increased expression of vascular endothelial growth factor (VEGF) and interferon inducible protein 10 (IP-10). Because of these three aforementioned unique anti-cancer potentials, OHSV-IL12 was tested in combination with local or systemic antiangiogenic inhibitors and immune checkpoint blockade. The combination of OHSV-IL12 + local angiogenic inhibitor produces anti-tumor effects by increasing intratumoral virus spread (as determined by X-gal staining for viral LacZ expression) and oncolysis, and by reducing CD31^+^ tumor vascularity and VEGF expression. The anti-tumor effects of the OHSV-IL12 + systemic angiogenic inhibitor are characterized by increased lysis of cancer cells and macrophage (CD68^+^) infiltration into tumors, increased apoptosis and necrosis in the tumor microenvironment, reduced tumor vascularity, and T cell dependent anti-tumor activity. OHSV-IL12 + immune checkpoint inhibitor produces robust and multifaceted anti-cancer activities, which include: oncolysis, increased infiltration of T cells and activated T cells into tumors, reduction of immunosuppressive regulatory T cells, increased effector (CD8^+^)/regulatory T cell (CD4^+^FoxP3^+^) ratio, Th1 differentiation, tumor-associated macrophage (TAM) infiltration and macrophage polarization towards M1-type, reduction of immune checkpoint PD-L1 positive cells, and induction of tumor-specific IFN-γ response. Upward and downward triangles indicate ‘increase’ and ‘decrease’, respectively.

**Figure 6 cells-09-00400-f006:**
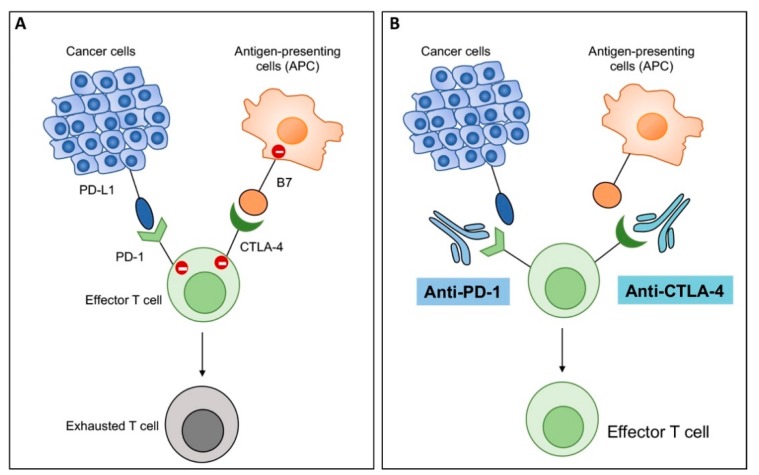
Graphic presentation of mechanism of action of immune checkpoint inhibitors (ICIs). (**A**) T cells express immune checkpoints programmed death 1 (PD-1) and cytotoxic T lymphocyte antigen 4 (CTLA-4), which interact with their corresponding ligands, i.e., programmed death ligand 1 (PD-L1) on cancer cells and B7 molecules on antigen presenting cells (APC), respectively. PD-1:PD-L1 and CTLA-4:B7 interactions send negative signals to immune cells, leading to exhaustion of T cells and no effector activity. (**B**) Antibodies to PD-1 (anti-PD-1) and CTLA-4 (anti-CTLA4) block those interactions and unleash anti-tumor immunity by enhancing activity of effector T cells.

**Table 1 cells-09-00400-t001:** List of OHSV-IL12s and their efficacy in pre-clinical cancer models.

Virus	Genomic Modification	Cancer Model	RoA	Dose (pfu)	Efficacy	Ref.
G47Δ-mIL12	ΔICP6, Δ∆ICP34.5, ΔICP47, ◦LacZ, ◦mIL-12	Intracranial 005 GSC (Glioblastoma)	I.T.	5 × 10^5^	Inhibited intracranial tumor growth and extended survival. Promoted IL-12 expression, stimulated IFN-γ production, upregulated IP-10, and inhibited VEGF. Polarized T_H_1 response and inhibited T-regs function.	[19,30]
T-mfIL12	ΔICP6, Δ∆ICP34.5, ΔICP47, ◦mIL-12	Intracerebral Neuro2a (neuroblastma)	I.V.	5 × 10^6^	Prolonged survival (Mock vs. T-mfIL12, *p* < 0.05), although not statistically significant versus T-01 treatments.	[31]
NV1042	ΔICP0, ΔICP4, ΔICP34.5, ΔUL56, ΔICP47, Us11Δ, Us10Δ, UL56 (duplicated), ◦mIL-12	Subcutaneous SCC VII (Squamous Cell Carcinoma)	I.T.	1 × 10^7^	Reduced tumor volume and improved survival (3 doses of 2 × 10^7^ pfu). 57% of mice from NV1042 group rejected subsequent SCC re-challenge in the contralateral flank compared with 14% in NV1023 or NV1034 group.	[32]
			I.V.	5 × 10^7^	NV1042 treatment resulted in 100% survival, in contrast to 70% of NV1023 and 0% of PBS.	[33]
M002	∆ΔICP34.5, ◦mIL-12	Intracranial X21415 (Pediatric embryonal tumor); D456 (pediatric glioblastoma); GBM-12 and UAB106 (adult glioblastoma)	I.T.	1 × 10^7^	M002 significantly prolonged survival in mice bearing all 4 types of tumor compared to saline. No difference in survival was observed compared with G207, excluding X21415 with high levels of nectin-1	[34]
		Intracranial SCK (brain metastasized breast cancer)	I.T.	1.5 × 10^7^	Single injection of M002 extended the survival of treated animals more effectively than a non-cytokine control virus.	[35]
		Xenograft SK-N-AS and SK-N-BE (2) (human neuroblastoma); subcutaneous Neuro-2a (murine neuroblastoma)	I.T.	1 × 10^7^	Significant decrease in tumor growth were observed in both SK-N-AS and SK-N-BE (2) cell lines. Extended median survival compared to the parent R3659.	[36]
		HuH6 (human hepatoblastoma; G401 (human malignant rhabdoid kidney tumor); SK-NEP-1 (renal Ewing sarcoma)	I.T.	1 × 10^7^	M002 significantly reduced tumor volume and increased survival over those treated with vehicle alone in all three different xenograft models.	[37]
R-115	Virulent with retargeted HER-2, ◦mIL-12	pLV-HER2- nectin-puro	I.P.	1 × 10^8^ to 2 × 10^9^	Induced greater local and systemic anti-tumor immunity and durable response than unarmed R-LM113 in both early and late schedule. All mice that survived from primary tumor challenge were protected from the distant challenge tumor and subsequent re-challenge. Increased number of CD8+ and CD141+ cells, PD-L1+ tumor cells, and Treg with a decrease in the number of CD11b+ cells. Enhanced Th1 polarization and increased expression of IFN-γ, IL-2, Granzyme B, T-bet and TNFα and tumor infiltrating lymphocytes	[38]
		Orthotopic mHGG^pdgf^- hHER2 (glioblastoma)	I.T.	Low dose: 2 × 10^6^ High dose: 1 × 10^8^	27% of mice treated with R-115 (*n* = 6, 4 low-dose arm and 2 high-dose arm) alive 100 days after the virus treatment versus all mice died within 48 days. Increased infiltrating CD4+ and CD8+, and expression of IFN-γ	[39]
vHSV-IL-12	ΔICP6, Δ∆ICP34.5, ◦mIL-12	Subcutaneous Neuro2a (neuroblastoma)	I.T.	1 × 10^4^	Significantly reduced tumor growth versus vHSV-null and other cytokine armed groups.	[40]
T2850	∆IR 15,091bp, ◦mIL-12	Subcutaneous A20 (Murine B Lymphoma), MC38 (colon adenocarcinoma), MFC (Murine Forestomach Carcinoma)	I.T.	1 × 10^7^	Reduced tumor volume compared to IL-12 unarmed parental group. IFN-γ level was markedly increased in the tumor bed and sera of mice infected with both T2850 and T3855 by day 4.	[41]
T3855	∆IR 15,091bp, ◦mIL-12, ◦mPD-1	Subcutaneous B16 (melanoma)		5 × 10^6^, 1 × 10^7^, 3 × 10^7^		
T3011	∆IR15,091bp, ◦hIL-12, ◦hPD-1	Subcutaneous B16 (melanoma)	I.T.	5 × 10^6^, 1 × 10^7^ or 3 × 10^7^	Reduced tumor volume as compared with control group.	[41]

∆—deletion, ◦ insertion, RoA—Route of administration, I.T—intratumorally, IP—intraperitoneally, pfu—plaque forming unit, ref.—reference, VEGF—vascular endothelial growth factor.

**Table 2 cells-09-00400-t002:** List of IL-12 expressing oncolytic viruses (other than OHSVs) and their efficacy in pre-clinical cancer models.

Virus	Strain	Cancer Model	RoA	Dose	Efficacy	Ref
Adenovirus	Ad5-yCD/ mutTKSR39 rep-mIL12	Subcutaneous TRAMP (C2 prostate adenocarcinoma)	I.T.	5 × 10^8^ pfu	Improved local and metastatic tumor control. Increased NK and CTL cytolytic activities. Significantly increased survival, levels of IL-12 and IFN-ƴ in serum and tumor.	[42,43]
	Ad-TD-IL-12, Ad-TD-nsIL-12	Subcutaneous HPD1NR (pancreatic cancer)	I.T.	1 × 10^9^ pfu	100% tumor eradication and survival of both IL-12 modified Adenovirus treated animals. Ad-TD-IL-12, but not Ad-TD-nsIL-12 resulted in a significant increase in CD3^+^CD4^−^CD8^+^ populations in the spleen. Level of splenic IFN-γ, IP-10 and lymph node IFN-γ were lower in Ad-TD-nsIL-12 compared to Ad-TD-IL-12 treated hamster.	[44]
	Ad-ΔB7/IL12/GMCSF	Subcutaneous B16-F10 (melanoma)	I.T.	5 × 10^7^ pfu	Primary tumor growth was better controlled in Ad-ΔB7/IL12/GMCSF and Ad-ΔB7/IL12 compared to Ad-ΔB7GMCSF or PBS. Increased tumor infiltrating CD86^+^ APCs and enhanced CD4+ and CD8+ T cell-mediated Th1 antitumor immune response.Reduced VEGF expression in the tumor treated with oncolytic Ad co-expressing IL-12 and GM-CSF or IL-12 alone. IFN-γ, TNF-α and IL-6 were higher in Ad-ΔB7/IL12/GMCSF and Ad-ΔB7/IL12 compared to Ad-ΔB7GMCSF or PBS.	[45]
	RdB/IL-12/ IL-18	Subcutaneous B16-F10 (melanoma)	I.T.	1 × 10^8^ pfu	95% and 99% tumor growth inhibition was observed in treatment with RdB/IL-12 and RdB/IL-12/IL-18, respectively. Increased Th1/Th2 cytokine ratio and increased tumor infiltration of CD4^+^ T, CD8^+^ T and NK cells. Promoted differentiation of T cells expressing IL-12Rβ2 or IL-18Rα.	[46]
	RdB/IL12/ DCN	Orthotopic 4T1 (Triple negative breast cancer)	I.T.	2 × 10^10^ VP	Both of the IL-12-expressing oncolytic Ads showed similar tumor growth inhibition up to day 9 after initial treatment. RdB/IL12/DCN increased upregulation of IFN-γ, TNF-α, infiltrating cytotoxic lymphocytes, downregulation of TGF-β expression and T-regs compared to RdB/IL12 and RdB/DCN.	[47]
	YKL-IL12/B7	Subcutaneous B16-F10 (melanoma)	I.T.	5 × 10^8^ pfu	Tumor growth was suppressed in both YKL-IL12 and YKL-IL12/B7 treated mice vs PBS. YKL-IL12- or YKL-IL12/B7-treated mice produced a significantly greater level of IFN-γ, infiltrating APCs, CD4+, CD8+ compared with PBS.	[48]
	Ad-ΔB7/ IL-12/4-1BBL	Subcutaneous B16-F10 (melanoma)	I.T.	5 × 10^9^ VP	100% of mice in the Ad-ΔB7/IL-12/4-1BBL group survived >30 days after initial viral injection compared with 20% of that in virus expressing either IL-12 or 4-1BBL. Mice treated with Ad-ΔB7/IL-12 or Ad-ΔB7/IL-12/4-1BBL had greater amount of tumor infiltrating CD4+ and CD8+ compared to Ad-ΔB7/4-1BBL and Ad-ΔB7.	[49]
Measles virus	MeVac FmIL-12	Subcutaneous MC38ce (colon carcinoma)	I.T.	5 × 10^5^–1× 10^6^ ciu	Tumor remissions in 90% of animals. Driven polarization of Th1-associated immune response and increased tumor infiltrating CD8^+^ T cells. Increased IFN-γ and TNF-α, and polarization of Th1-associated immune response. Co-expression of IL-12 and IL-15 showed synergistic effect.	[20,50]
Maraba Virus	MG1-IL12- ICV	CT26 and B16F10 peritoneal carcinomatosis	I.P	Seeding dose 5 × 10^5^, then 1 × 10^4^ on day 3	Reduced tumor burden and improved mouse survival. Activated and matured DCs to secrete IP-10, and activated and recruited NK cells. Increased production of IFN-γ	[51]
Newcastle disease virus	rClone30– IL-12	Orthotopic H22 (hepatocarcinoma)	I.T.	1 × 10^7^ pfu	Reduced tumor volume and improved percentage of survival. Increased IFN-γ and IP-10. Co-expression of IL-12 and IL-2 showed synergistic effect.	[52]
Semliki Forest virus	rSFV/IL12	Subcutaneous B16 (melanoma)	I.T.	10^7^ IU	Single injection with SFV-IL12 resulted in significant tumor regression. 2 days after injection, IFN-γ production increased with inhibition of tumor vascularization. Splenic IP-10 and MIG expression was increased.	[53]
	SFV/IL12	Subcutaneous P815 (mastocytoma)	I.T.	10^6^ IU	Significantly delayed P815 tumor growth. 40–53% of mice exhibited complete tumor regressions. Induced high levels of IFN-γ production in draining lymph nodes.	[54]
	SFV/IL12	Subcutaneous MC38 (colon adenocarcinoma)	I.T.	10^8^ particles	Reduced tumor volume and improved percentage of survival. Increased tumor-specific CD8+ T lymphocytes. Enhanced the expression of CD11c, CD8α, CD40, and CD86 of tumor-infiltrating M-MDSCs in the presence of an intact endogenous IFN-I system.	[55]
	SFV-VLP-	Syngeneic RG2 (rat glioma)	I.T.	5 × 10^7^ (low-dose) or 5 × 10^8^ (high-dose)	Reduction in tumor volume (70%—low dose; 87%—high dose)	[56]
Vesicular stomatitis virus (VSV)	VSV-IL12	Orthotopic SCC VII (squamous cell Carcinoma)	I.T.	MOI 0.01	Significant reduction in tumor volume, and prolonged survival.	[57]
Sindbis virus	Sin/IL12	Orthotopic ES-2 cells (ovarian clear cell Carcinoma)	I.P	10^7^ pfu	Reduced tumor growth and improved survival. Activated and matured DCs, activated and recruited NK cells. Increased production of IFN-γ.	[58]

∆—deletion, RoA—Route of administration, I.T—intratumorally, IP—intraperitoneally, pfu—plaque forming unit, ref. —reference, MOI—multiplicity of infection, VP—viral particle, IU—infectious units.

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
