# Peer review of "Oncolytic Virus Encoding a Master Pro-Inflammatory Cytokine Interleukin 12 in Cancer Immunotherapy"

_cells, 2020, doi:10.3390/cells9020400_

Round 1

Reviewer 1 Report

The authors have done a enormous summary of bibliography on OHSV-IL12. To my knowledge, this type of review has not been done before. We mostly find review for a recombinant OHSV or an application model but no exhaustive review of application to all type of cancers.

The problem is that, in some paragrap, and particularly in paragraph 2.1, exemples of recombinant virus are listed wich renders the text hard to follow. The authors compare different types of recombinant OHSV and applications without presenting a figure of gene and sequence of HSV. For people not aware of HSV sequence, the paragraph2.1  is incomprehensible.

Therefore I would recommand to add a simple cartoon of HSV sequence encompassing the deleted and modified genes. 

The other paragraphs are well structured and easier to follow. It gives very informative comparison of combination and mechanism of action of existing oncolytic virotherapy under development, their limitations and current optimisation.

Maybe an additionnal figure on ICI and their ligand/mechanism of action could be a plus for non immunologist in paragraph 2.4.

Author Response

We sincerely thank and appreciate the reviewer for his/her time and effort in reviewing this manuscript. Please find below point-by-point responses to reviewer’s comments.

Reviewer 1

The authors have done a enormous summary of bibliography on OHSV-IL12. To my knowledge, this type of review has not been done before. We mostly find review for a recombinant OHSV or an application model but no exhaustive review of application to all type of cancers.

We truly appreciate your comments.

The problem is that, in some paragraph, and particularly in paragraph 2.1, examples of recombinant virus are listed which renders the text hard to follow. The authors compare different types of recombinant OHSV and applications without presenting a figure of gene and sequence of HSV. For people not aware of HSV sequence, the paragraph 2.1 is incomprehensible. Therefore, I would recommend to add a simple cartoon of HSV sequence encompassing the deleted and modified genes.

Thank you for this comment. We have added a new figure (read as Figure 2) which represents HSV genes that were deleted and/or modified during construction of engineered OHSV-IL12 viruses.   

The other paragraphs are well structured and easier to follow. It gives very informative comparison of combination and mechanism of action of existing oncolytic virotherapy under development, their limitations and current optimization.

Thank you.

Maybe an additionnal figure on ICI and their ligand/mechanism of action could be a plus for non-immunologist in paragraph 2.4.

We have added a new figure (read as Figure 6). Thank you.

Reviewer 2 Report

This review of the oncolytic virus encoding a pro-inflammatory master cytokine, interleukin 12, in cancer immunotherapy is divided into 8 chapters. First, it describes the major human oncolytic herpesviruses that have been developed and their use in the treatment of model tumours. A brief overview of other oncolytic viruses expressing IL12 and their application in cancer treatment is provided at the end of the review.
In my opinion, this review is well written and very informative, with a focus on a specific topic. Figures 1 to 3 help in understanding the main topic and description of oncolytic viruses. They are more devoted to the first part of the manuscript. Figure 4 is probably less easy to read without comments. It would probably be useful to add a more detailed description to the legend. Some of the abbreviations used are not described, which may make clarity more difficult for non-specialists. I assume that GBM means Glioblastoma and GSC glioblastoma stem like cells! What about SCC?
Page 2 line 57 to 60, the reference to Table II should be placed here and the bibliographic references should be added.
Page 12 line 409: I do not understand the reference to Figure 3.
Line 426: The reference to Table II should be added.

Author Response

We sincerely thank and appreciate the reviewer for his/her time and effort in reviewing this manuscript. Please find below point-by-point responses to reviewer’s comments.

This review of the oncolytic virus encoding a pro-inflammatory master cytokine, interleukin 12, in cancer immunotherapy is divided into 8 chapters. First, it describes the major human oncolytic herpesviruses that have been developed and their use in the treatment of model tumours. A brief overview of other oncolytic viruses expressing IL12 and their application in cancer treatment is provided at the end of the review. In my opinion, this review is well written and very informative, with a focus on a specific topic. Figures 1 to 3 help in understanding the main topic and description of oncolytic viruses.

          We thank you for expressing your valuable opinion.

They are more devoted to the first part of the manuscript.

           Yes, our major goal was to focus on oncolytic HSVs. However, in this revised version we have expanded our text to other types of OV as well (please find track changes in section 3).

Figure 4 is probably less easy to read without comments. It would probably be useful to add a more detailed description to the legend.

            We have provided a detailed description in the legend, which should now make this figure easier to follow (please read this figure as Figure 5 in the revised version of the manuscript).

Some of the abbreviations used are not described, which may make clarity more difficult for non-specialists. I assume that GBM means Glioblastoma and GSC glioblastoma stem like cells! What about SCC?

           We apologize for the lack of clarification. We have now provided a description of each abbreviation in appropriated places in the text. SCC stands for squamous cell carcinoma (please see track changes).    

Page 2 line 57 to 60, the reference to Table II should be placed here and the bibliographic references should be added. 

            We have referenced Table 2 (now lines 67-71).

Page 12 line 409: I do not understand the reference to Figure 3. 

We apologize for this typo. It should have been Figure 4 in our previous version of the manuscript. In the revised text it is Figure 5.

Line 426: The reference to Table II should be added.

            Added. Thank you.

Reviewer 3 Report

Nguyen et al, have described developments in proinflammatory cytokine, IL-12-based oncolytic virotherapy. They have widely covered about IL-2 antitumor activity; engineering strategies with HSV and other viruses; antitumor efficacy in preclinical as well as clinical studies. This manuscript is very interesting, however there are some issues as follows:

Authors are mostly focused on HSV. In the table, Adenovirus was mentioned though, how about other type of virus? I am interested in other type of virus’s studies also. Would you add the cases of other types of engineered OV expressing IL12 and their outcomes in the Table and Text content together? (Topic 3) Authors are recommended to include the potential of IL12, compared to other cytokines. The mechanism, IL12 itself vs. OV-IL12 should be described. Why Oncolytic virus is needed to enhance the potential of IL12 while avoiding the side effect of IL12 itself? What is pre- or clinical status of each engineered OV-IL12. Is there any cancer type preference? Combination therapy was introduced, but not focused well. Authors recommended to discuss regarding this. Also, clinical perspective is very short and look like stop writing at the end.

Minor points:

Line 10-11 “OVs are …… induce anti-tumor immunity”: This statement is partially true, the reason is some of the natural viruses are also used as OVs in oncolytic viraltherapy. Please reflect the fact.

Topic 2. Safety and anti-tumor efficacy of IL-12 expressing OHSVs: This topic widely describes engineering strategies and therapeutic efficacy of IL-2 encoded HSV, or an alternative title can be given here (e.g. HSVs engineering with IL-2 and their therapeutic efficacy etc.)

Line 181-182. Please revise the sentence.

Line#249: Depletion here; not deletion

Author Response

We sincerely thank and appreciate the reviewer for his/her time and effort in reviewing this manuscript. Please find below point-by-point responses to reviewer’s comments.

Reviewer 3

Nguyen et al, have described developments in proinflammatory cytokine, IL-12-based oncolytic virotherapy. They have widely covered about IL-2 antitumor activity; engineering strategies with HSV and other viruses; antitumor efficacy in preclinical as well as clinical studies. This manuscript is very interesting, however there are some issues as follows:

We are happy to hear that you found our manuscript interesting. Please find below point-by-point responses to your comments.

Authors are mostly focused on HSV. In the table, Adenovirus was mentioned though, how about other type of virus? I am interested in other type of virus’s studies also. Would you add the cases of other types of engineered OV expressing IL12 and their outcomes in the Table and Text content together? (Topic 3)

Thank you for this comment. Since oncolytic HSV (OHSV) is the furthest along in the clinic and OHSV is the only oncolytic virus approved by the FDA for the treatment of cancer in the USA at this time, our discussion in this manuscript was mainly focused on HSV. However, we understand that other viruses are equally valuable and important in the field of cancer therapy. Therefore, as per your suggestion, we have expanded the text in section 3, which now presents important studies with other types of OV expressing IL-12.

Authors are recommended to include the potential of IL12, compared to other cytokines. The mechanism, IL12 itself vs. OV-IL12 should be described.

We have included therapeutic potential and mechanisms of action of IL-12 in the first paragraph of the Introduction section and the mechanisms of OV-IL12 in Figure 1. The goal of this manuscript is to show how IL-12 was utilized to enhance OV-induced anti-tumor effects (such as anti-tumor immunity and anti-angiogenesis, as presented in the Figure 1) rather than to compare and contrast IL-12 with other cytokines. However, a separate manuscript is currently under preparation describing the anti-cancer potential of different immunostimulatory cytokines/chemokines, including IL-12.

Why Oncolytic virus is needed to enhance the potential of IL12 while avoiding the side effect of IL12 itself?

We are not quite clear about this statement and we apologize for the confusion. IL-12 is inserted in the viral genome to improve the OV-induced vaccine effects, not the other way around (please see lines 62-64). Engineered OV-IL12 releases IL-12 locally in the tumor, thus avoiding toxicities associated with systemic IL-12 administration.

What is pre- or clinical status of each engineered OV-IL12.

For preclinical studies, please refer to Table 1 and Table 2, and the manuscript text. There are only a few studies are currently under clinical trial evaluation, which we have included in the section 4 (i.e., clinical perspectives).

Is there any cancer type preference? Combination therapy was introduced, but not focused well. Authors recommended to discuss regarding this. Also, clinical perspective is very short and look like stop writing at the end.

There is no cancer type preference, since OV-IL12 has been tested and found effective in various cancers, such as glioma, neuroblastoma, squamous cell carcinoma, metastatic breast cancer, hepatoblastoma, sarcoma, kidney cancer, lymphoma, prostate cancer, pancreatic cancer, colon cancer, ovarian cancer, melanoma, etc. We have included above statement in the manuscript text.

Manuscript text in sections 2.4 and 2.5 are entirely combination studies. In addition, we have incorporated new studies in section 3 that also include therapeutic effects of virally expressed IL12 in combination with other virally expressed cytokine/co-stimulatory ligands. OV-IL12 based treatment is a growing field and there are not many published pre-clinical combination studies available in the literature. Similarly, very few clinical studies are currently under evaluation, which we have incorporated in the “Clinical Perspectives” section.

Minor points:

Line 10-11 “OVs are …… induce anti-tumor immunity”: This statement is partially true. The reason is some of the natural viruses are also used as OVs in oncolytic viral therapy. Please reflect the fact.

            Corrected our statement. Thank you.

Topic 2. Safety and anti-tumor efficacy of IL-12 expressing OHSVs: This topic widely describes engineering strategies and therapeutic efficacy of IL-2 encoded HSV, or an alternative title can be given here (e.g. HSVs engineering with IL-2 and their therapeutic efficacy etc.)

            An alternative title has been given. Thank you.

Line 181-182. Please revise the sentence.

           Revised (now it is line 222).

Line#249: Depletion here; not deletion

            Corrected (now it is in line 289). Thank you.

Round 2

Reviewer 3 Report

Still feel like to hear more about the mechanism behind and why IL12 is better than other cytokines, though; this manuscript is educative in terms of learning OV-IL12s. Thank you for the work.